# *OsBTBZ1* Confers Salt Stress Tolerance in *Arabidopsis thaliana*

**DOI:** 10.3390/ijms241914483

**Published:** 2023-09-23

**Authors:** Triono B. Saputro, Bello H. Jakada, Panita Chutimanukul, Luca Comai, Teerapong Buaboocha, Supachitra Chadchawan

**Affiliations:** 1Center of Excellence in Environment and Plant Physiology, Department of Botany, Faculty of Science, Chulalongkorn University, Bangkok 10330, Thailand; trionobsaputro@gmail.com (T.B.S.); bellojakada@gmail.com (B.H.J.); 2Program in Biotechnology, Faculty of Science, Chulalongkorn University, Bangkok 10330, Thailand; 3National Center for Genetic Engineering and Biotechnology, National Science and Technology Development Agency, Khlong Luang, Pathumthani, Bangkok 12120, Thailand; priggerr@gmail.com; 4Genome Center and Department of Plant Biology, UC Davis, Davis, CA 95616, USA; lcomai@ucdavis.edu; 5Center of Excellence in Molecular Crop, Department of Biochemistry, Faculty of Science, Chulalongkorn University, Bangkok 10330, Thailand; teerapong.b@chula.ac.th; 6Omics Science and Bioinformatics Center, Faculty of Science, Chulalongkorn University, Bangkok 10330, Thailand

**Keywords:** abiotic stress, abscisic acid, BTB domain, BTBZ, salt stress, tolerance

## Abstract

Rice (*Oryza sativa* L.), one of the most important commodities and a primary food source worldwide, can be affected by adverse environmental factors. The chromosome segment substitution line 16 (CSSL16) of rice is considered salt-tolerant. A comparison of the transcriptomic data of the CSSL16 line under normal and salt stress conditions revealed 511 differentially expressed sequence (DEseq) genes at the seedling stage, 520 DEseq genes in the secondary leaves, and 584 DEseq genes in the flag leaves at the booting stage. Four *BTB* genes, *OsBTBZ1*, *OsBTBZ2*, *OsBTBN3*, and *OsBTBN7*, were differentially expressed under salt stress. Interestingly, only *OsBTBZ1* was differentially expressed at the seedling stage, whereas the other genes were differentially expressed at the booting stage. Based on the STRING database, *OsBTBZ1* was more closely associated with other abiotic stress-related proteins than other *BTB* genes. The highest expression of *OsBTBZ1* was observed in the sheaths of young leaves. The OsBTBZ1-GFP fusion protein was localized to the nucleus, supporting the hypothesis of a transcriptionally regulatory role for this protein. The *bt3 Arabidopsis* mutant line exhibited susceptibility to NaCl and abscisic acid (ABA) but not to mannitol. NaCl and ABA decreased the germination rate and growth of the mutant lines. Moreover, the ectopic expression of *OsBTBZ1* rescued the phenotypes of the *bt3* mutant line and enhanced the growth of wild-type *Arabidopsis* under stress conditions. These results suggest that *OsBTBZ1* is a salt-tolerant gene functioning in ABA-dependent pathways.

## 1. Introduction

Rice (*Oryza sativa* L.) is one of the most important primary food resources worldwide. In Asia, rice production is a principal factor for improving food security. Salinity is a major limiting factor for plants such as rice, decreasing their growth and productivity [1]. Moreover, salt toxicity adversely affects the grain yield, panicle length, spikelet number per panicle, seed weight per panicle, and 1000-grain weight [2,3]. Soil salinity can occur naturally or can be induced by human activities, such as constant irrigation with low-quality groundwater [4]. High salt concentrations can adversely affect plant physiology through ion toxicity in the plant cells, which reduces the photosynthesis rate and growth of plants [5].

Salt stress tolerance is a polygenic trait controlled by multiple genes in the rice genome. Various efforts have been implemented to determine the genes or genomic regions responsible for this trait. Molecular markers for salt-tolerant phenotypes have been identified by several research groups [6,7,8,9,10,11,12,13,14,15,16]. After the development of the omics sciences, genomics, transcriptomics, and proteomic approaches have been used to identify the genes/proteins involved in salt tolerance, including their functions in ion transport regulation [17]. A greater understanding of the salt tolerance mechanisms initiated by other genes has also been elucidated. Recently, the negative regulator [18] and protein with a role in absorbed light energy dissipation [19] were reported to have a role in salt tolerance.

A genome-wide association study (GWAS) was conducted on the salt-tolerance traits in rice at the germination [20,21], seedling [18,22,23,24], early vegetative [25,26], and flowering stages [27,28]. In response to various environmental stress conditions, plants alter their gene expression to deal with the negative effects of environmental signals. Transcriptomics is a prominent method for identifying the genes that potentially regulate salt tolerance in rice [29]. Transcriptomics analysis can be used in combination with genomic data to predict salt tolerance genes in rice. Lv et al. [30] performed a GWAS using 3.82 million SNPs associated with the standard evaluation score (SES) of visual salt injury and then combined them with the differentially expressed genes between cultivars 93-11 and PA64s under normal and salinity stress conditions to predict 30 candidate salt-tolerant genes.

The chromosome segment substitution line 16 (CSSL16) is a salt-tolerant line. Based on the DEseq data of CSSL16, along with the genomic comparison between CSSL16 and its original genomic background, ‘Khao Dawk Mali 105 (KDML105)’, Chutimanukul et al. [31] analyzed the transcriptome data of ‘KDML105’ and CSSL16 rice, using a gene co-expression network (GCN), a weighted gene co-expression network (WGCN), and clustering analysis and predicted 92 candidate salt-tolerant genes. Then, this information was combined with a genomic comparison between CSSL16 and KDML105 and revealed nine candidate genes, seven of which were in the salt-tolerant QTL previously reported by Kanjoo et al. [32,33]. In this research, we report the validation of the *OsBTBZ1* gene, which is one of the nine candidate salt-tolerant genes predicted by the combined methods of transcriptomic analysis via GCN, WGCN, and CC and genomic comparison.

*OsBTBZ1* (*LOC_Os01g66890*) was predicted as an important gene responsible for the salt tolerance characteristics of CSSL16 [31]. It is a Bric-a-Brac, Tramtrack, and Broad Complex BTB domain with a TAZ zinc finger and Calmodulin-binding domains. BTB proteins have been studied for many crops and play various roles, mainly in plant growth and in responses to abiotic stimuli. For example, the expression of the *CsBT1* gene in cucumber plants notably decreased under salt stress [34], whereas the *CaBPM4* gene in pepper was induced after 8 h under salt and drought exposure and 12 h after exposure to cold stress [35]. Moreover, the expression in *Arabidopsis thaliana* of the *IbBT4* gene from sweet potato enhanced drought tolerance [36]. 

In this research, to understand the function of *OsBTBZ1* in salt stress conditions, the *OsBTB* gene family expression at both the seedling and booting stages was investigated. Phylogenetic analysis of the *OsBTB* gene family was performed. The cis-elements in the promoters of *OsBTB* genes induced by salt stress were compared to support the salt-responsive expression of the genes. Based on the amino acid sequence of the OsBTBZ1 protein, it is predicted to be a transcription factor and to regulate other genes of the salt-tolerant phenotype [31]. This gene was reported to be involved in plant growth regulation; however, its function during salt exposure has not yet been characterized fully. *AtBT3* is the *OsBTBZ1* ortholog in *Arabidopsis*. The *Atbtbz1* (*Atbt3*) mutant, a null mutant of the *AtBT3* gene, is more susceptible to salt stress [31]. In this study, the *OsBTBZ1* expression cassette was transferred to *Arabidopsis* wild-type (WT) plants for ectopic expression and to the *Atbt3* mutant for a complementation study. The homozygous T_3_ plants were used to investigate the salt, ABA, and mannitol responses and clarify the function of the OsBTBZ1 protein in these abiotic stresses.

## 2. Results

### 2.1. Only Four BTB Genes Were Expressed in CSSL16 under Salt Stress 

In order to investigate whether all *OsBTB* genes can be induced by salt stress, 182 genes containing the BTB domains were retrieved from the Phytozome database [37], and the differentially expressed genes at seedlings and booting stages were explored. These two stages, the seedling and booting stages, of rice were selected for transcriptome analysis because they are susceptible to salt stress [18,28]. Moreover, the flag leaves and second leaves of rice at the booting stage are important for grain-filling. Under salt stress, 511 genes were differentially expressed at the seedling stage, whereas 520 and 584 differentially expressed genes were found in the second and flag leaves, respectively. Among them, only four *BTB* genes, *LOC_Os01g66890* (*OsBTBZ1*), *LOC_Os01g68020* (*OsBTBZ2*), *LOC_Os02g38120* (*OsBTBN3*), and *LOC_Os03g41350* (*OsBTBN7*), were differentially expressed in salt-stressed CSSL16, while *OsBTBZ1* was the only *BTB* gene expressed at the seedling stage, and the other genes were expressed at the booting stage in both second and flag leaves (Figure 1A). Furthermore, OsBTBZ1 and OsBTBZ2 contained the BTB and Transcription Adaptor putative zinc finger (TAZ zF or zF-TAZ) domains, whereas OsBTBN3 and OsBTBN7 contained the non-phototropic hypocotyl3 (NPH3) domain (Figure 1B). 

Phylogenetic analysis of the genes containing the BTB domain in rice and *Arabidopsis* was conducted using the JTT model, which used a substitution model calculated from the nearest-neighbor proteins with more than 85% similarity. OsBTBZ1 and its ortholog in *Arabidopsis* AT1G05690 (AtBT3) belong to the same cluster. Both OsBTBZ1 and OsBTBZ2 are in the same cluster and contain the zf-TAZ domain. Moreover, OsBTBN3 and OsBTBN7, which contain an NPH3 domain, were clustered together (Figure 2A). The similar domains in OsBTBZs and OsBTBNs suggest a similar function for these proteins.

The salt-tolerant quantitative trait locus (QTL) identified on chromosome 1 and located between RM1003 and RM3362 contained four *BTB* genes: *OsBTBZ1*, *OsBTBZ2*, *OsBTBM1* (*LOC_Os01g70670*), and *OsBTBA3* (*LOC_Os01g72020*). *OsBTBN3* and *OsBTBN7* were located on chromosomes 2 and 3, respectively. Furthermore, chromosomes 8 and 10 contained dense clusters of *BTB* genes, with 34 and 46 *BTB* genes, respectively (Figure 2B).

### 2.2. OsBTBZ1, OsBTBZ2, OsBTBN3, and OsBTBN7 Promoters Contain Multiple Cis-Elements Related to the Water Stress Response

An investigation of the cis-element at the promoter regions of *OsBTBZ1*, *OsBTBZ2*, *OsBTBN3*, and *OsBTBN7* was performed to reveal the salt-responsive elements located in these *OsBTB* genes. Two thousand base pair sequences upstream of the *OsBTBZ1*, *OsBTBZ2*, *OsBTBN3*, and *OsBTBN7* transcription start sites were analyzed using New PLACE to identify the regulatory cis-elements located in the promoter regions of these genes (Figure 3). Table 1 summarizes the cis-elements related to abiotic stress. Multiple MYCCONSENSUSAT elements, which are water stress-responsive elements, were found in all the tested promoters. More than 10 of these elements were located on the *OsBTBZ1*, *OsBTBZ2*, and *OsBTBN3* promoters, which is consistent with the upregulation pattern under salt stress conditions. Moreover, other cis-elements related to dehydration stress, such as ACGTATERD1, DPBFCOREDCDC3, and MYBCORE, were found in all four promoters. Interestingly, ABA-responsive elements were found only in the + strand of *OsBTBZ1*. The GAGA-binding protein binding site (GAGAGMGSA1), which is specific to the CSSL16 allele, but not the ‘KDML105′ allele, was detected only in *OsBTBZ1* (+ strand) (Table 1 and Figure 3).

### 2.3. OsBTBZ1 and OsBTBZ2 Are in the Same Protein-Protein Interaction (PPI) Network

The STRING database was used to investigate the protein-protein interaction (PPI) network of the four BTB proteins: OsBTBZ1, OsBTBZ2, OsBTBN3, and OsBTBN7 to select the best candidate OsBTB protein, which showed the highest number of connections with other proteins, for further characterization. OsBTBZ1 and OsBTBZ2 were connected, whereas OsBTBN3 and OsBTBN7 were dissociated. Furthermore, OsBTBZ1 and OsBTBZ2 were linked to other proteins, including CARM1, OS07T0626600-01, GCN5, and OS01T0884500-01. CARM1, OS07T0626600-01, and GCN5 are involved in chromatin remodeling. Modifying the chromatin architecture is necessary for the epigenetic control of gene expression, which does not involve alterations in DNA sequences, while increased chromatin compaction results in distinct higher-order structures [38]. OS01T0884500-01 is a zP (CCCH-type) protein. Contrastingly, OsBTBZ1 is connected to OS03T0216600-01, a glucan 1,3-alpha-glucosidase, and is also connected with OsJ_06167, OsWRKY39, and OsJ_25984, a protein kinase [39], which mediates pathogen-associated molecular pattern (PAMP)-triggered responses (Figure 4).

OsBTBN7 was associated with three uncharacterized proteins: OS03T0695500-01, OS04T0258900-01, and OS04T0283600-00, whereas OsBTBN3 was not associated with any other protein (Figure 4). Therefore, the functions of these two proteins were not investigated further.

According to the PPI network prediction using STRING, OsBTBZ1 was predicted to interact with more proteins. Therefore, it was selected for further characterization.

### 2.4. OsBTBZ1 Is Expressed in All Plant Tissues, Especially in Younger Leaf Sheaths

Because *OsBTBZ1* showed differential expression owing to salt stress at the seedling stage, 15-day-old seedlings were selected for monitoring the expression of this gene. High gene expression was detected in the young leaves and root tissues, and a relatively higher level of expression was detected in the young (first leaf) leaf sheath, compared to the young leaf blades. However, the expression levels decreased in older leaves. In the oldest leaf of the 15-day-old seedlings, *OsBTBZ1* expression was lower in the leaf sheath than in the leaf blade. At the reproductive stage, *OsBTBZ1* was highly expressed in the leaf sheaths of flag leaves, as well as in the peduncles and spikelets (Figure 5). 

### 2.5. OsBTBZ1 Is Localized in the Nucleus, Suggesting the Role of the Transcription Factor

To analyze the subcellular localization of the OsBTBZ1 protein, the CDS of *OsBTBZ1* was fused with *GFP* and transiently expressed in the epidermal cells of *Allium cepa*. *OsBTBZ1-GFP* expression resulted in fluorescence in the nucleus, whereas the GFP fluorescence of the control was observed outside the nuclear region (Figure 6). *IbBT4*, a *BTB* gene in *Ipomoea*, is also localized in the nucleus [36]. A similar result was reported by Weber and Hellmann [40], who examined the BPM1, BPM2, and BTB proteins in *Arabidopsis*. This is consistent with a transcription factor role for OsBTBZ1, which may regulate other genes that are responsible for salt tolerance.

### 2.6. Ectopic Expression of OsBTBZ1 Could Revert the NaCl and ABA Susceptibility of the Atbt3 Arabidopsis Mutant at the Germination Stage

Chutimanukul et al. [31] reported that the *Atbt3 Arabidopsis* mutant was more susceptible to salt stress. Therefore, to investigate the role of *OsBTBZ1* an ortholog to *AtBT3*, we fused it to the constitutive *CaMV35S* promoter and transformed it into the WT *Arabidopsis* and *Atbt3* mutant. We generated two lines (REV1 and REV2) in the *Atbt3* mutant background to test for complementation of salt sensitivity, along with two lines, OE1 and OE2, in the WT background to test for the effect of overexpression.

Germination tests were performed to investigate the effects of NaCl, ABA, and mannitol stress on *Arabidopsis* seed germination. The responses are shown in Figure 7A–E. Under normal conditions, the seeds of all lines, comprising the WT, *Atbt3* mutant (*bt3*), REV1, REV2, OE1, and OE2, displayed no significant difference in germination rate (Figure 7A,B).

Expression of *OsBTBZ1* could enhance the germination rate of *bt3* mutant lines. When 150 mM NaCl was applied to 0.5× MS medium, the WT germination percentage was reduced to 88%, whereas the *bt3* mutant line showed only 54% germination after 6 d of germination. The ectopic expression of *OsBTBZ1* in the *bt3* mutants reversed this effect. Both REV1 and REV2 had germination rates of 70% after 9 d of germination. Moreover, the OE1 and OE2 lines, which expressed *OsBTBZ1* in the WT, had a germination rate similar to that of the WT (Figure 7A,C).

The application of ABA to the medium delayed seed germination in all lines. On 0.5× MS medium supplemented with ABA, no germination occurred in the WT, *bt3* mutant, and REV2 after 4 d of germination (Figure 7A,D). Later, after 5 d of incubation, the germination rate of all lines increased and reached a maximum after 10 d. A 100% germination rate was detected in REV1 and OE2, while WT showed an 86% germination rate. The lowest germination rate under ABA was found in the *bt3* mutant (73%). The ectopic expression of *OsBTBZ1* could revert ABA susceptibility in the *bt3* mutant line by increasing the germination rate to over 80% in both REV1 and REV2. In the WT background, *OsBTBZ1* expression enhanced germination, and, after 5 d of germination, approximately 40% germination was detected in both OE1 and OE2, whereas WT showed a germination rate of <20%. Although WT and OE1 showed similar levels of germination (73%) after 10 d of germination, OE2 showed 100% germination (Figure 7A,D).

To discern the precise function of the *OsBTBZ1* gene, it is essential to evaluate its role in germination under mannitol-induced conditions, which will help in determining whether it is solely responsive to salt or has a broader function within the osmotic regulation mechanisms. Therefore, 150 mM mannitol was added to the MS medium as a germination test. A slight reduction in the WT seed germination percentage was detected. The mutation of *AtBt3* increased the sensitivity to mannitol stress, as shown by the significantly lower germination percentage of the *Atbt3* mutant. However, the ectopic expression of *OsBTBZ1* did not reverse the inhibition of mannitol-induced germination (Figure 7A,B,E).

Based on these results, we can conclude that *OsBTBZ1* can reverse the *Atbt3* mutation under salt and ABA stress, but not under drought stress that is induced by mannitol.

### 2.7. OsBTBZ1 Enhanced the Salt and ABA Tolerance in Transgenic Arabidopsis

To investigate whether *OsBTBZ1* enhanced the stress tolerance of plants, the fresh weight, root length, and photosynthetic pigment content under stress conditions were measured. At the beginning of the experiment (0 d), seedlings of the WT, *bt3* mutant, REV1, REV2, OE1, and OE2 lines showed similar fresh weights (Appendix A). After 6 d of the experiment, the *bt3* mutant had a significantly lower fresh weight than the WT. Ectopic expression could reverse the phenotype of the *bt3* mutant to a fresh weight similar to that of the WT under normal conditions, as shown by the phenotypes of REV1 and REV2 (Figure 8A,B). Moreover, the root lengths of all the lines were similar under normal conditions (Figure 8A,C).

Salt stress and ABA treatments reduced the fresh weight and root length of all lines (Figure 8D–I). The fresh weight of WT was reduced by 50%, whereas that of the *Atbt3* mutant decreased by more than 60%. This indicated that the *Atbt3* mutant was more susceptible to salt stress. The revertant lines, REV1 and REV2, could reverse NaCl susceptibility by showing a significantly higher fresh weight than the *Atbt3* mutant after 6 d of the NaCl treatment. OE1 and OE2 also had higher fresh weights than WT (Figure 8E). A similar response could also be detected via the root-length response under salt stress (Figure 9F). The effect of 1 mM of ABA on *Atbt3* growth was not as strong as that of the NaCl treatment. Approximately 50% of the fresh weight was reduced in the ABA-treated *Atbt3* mutant after 6 d, whereas a similar reduction in the root length of the WT and mutant was detected. The ectopic expression of *OsBTBZ1* in both the WT and mutant backgrounds reversed these effects (Figure 8G–I).

Mannitol reduced the fresh weight and root length of all lines; however, no significant difference in fresh weight was detected (Figure 8J–L). Conversely, the ectopic expression of *OsBTBZ1* enhanced the root length (Figure 8L).

We also compared the photosynthetic pigment levels. On the first day of the experiment, all lines had similar levels (Appendix A). Under normal conditions, after 6 d, *OsBTBZ1* ectopic expression significantly enhanced the Chl *b* content in the *Atbt3* mutant, showing a tendency to enhance the Chl *a* and carotenoid contents in the WT (Figure 9A–C). 

Salt stress (150 mM NaCl) reduced the Chl *a*, Chl *b*, and carotenoid contents in all lines. Strong reductions in the Chl *a* and carotenoid contents were detected in the *Atbt3* mutant. However, these effects were reversed by *OsBTBZ1* expression, as shown by the REV1 and REV2 pigment contents. The ectopic expression of *OsBTBZ1* in the WT did not show significant levels of Chl *a*, Chl *b*, and carotenoid contents when compared to the pigment content of the WT (Figure 8D–F). 

ABA significantly decreased the Chl *a*, Chl *b*, and carotenoid contents in all lines (Figure 8G–I), with a much stronger effect on the Chl *b* content (Figure 8B,E,F). The *Atbt3* mutant was more susceptible to ABA treatment in terms of the photosynthetic pigment content. An approximately 60–75% reduction in Chl *a* and Chl *b* content was detected in the *Atbt3* mutant, respectively, after ABA treatment, whereas in the WT, an approximately 40–60% reduction in Chl *a* and Chl *b* contents was found (Figure 8G,H), along with a reduction in the carotenoid content (Figure 8I). The ectopic expression of *OsBTBZ1* reversed these effects.

Furthermore, treatment with 150 mM of mannitol caused a similar reduction in the photosynthetic pigment content in all lines (Figure 8J–L).

Based on the earlier investigation into growth and photosynthetic pigment contents, the expression of *OsBTBZ1* in the WT and *Atbt3* mutant background could confer abiotic stress tolerance under salt and ABA stress and showed fewer effects under drought stress when treated with 150 mM of mannitol.

## 3. Discussion

The BTB domain genes are part of a large gene family that has various roles in plant responses to abiotic stresses, ubiquitination, and development. In the present study, *OsBTBZ1*, *OsBTBZ2*, *OsBTBN3*, and *OsBTBN7* were induced by salt stress at various stages, suggesting roles in the salt stress response in both the seedling and booting stages. The *BTB* genes in other species have also been reported to be involved in salt stress. For example, the *A. thaliana* stress-induced BTB protein 1 (AtSIBP1) is a positive regulator of salinity responses in *Arabidopsis* [41]. CaBPM4 (*Capsicum annuum* BTB-POZ and MATH domain protein) from pepper is upregulated during salt-stress exposure [35]. The SlBTB18 in tomatoes contains the TAZ domain and its expression increases dramatically under cold, salt, and oxidative stress [42]. The *AtBt3* gene, which was in the same cluster as *OsBTBZ1*, has been reported to play a crucial role in gametophyte development in *Arabidopsis* [43]. Chutimanukul et al. [31] demonstrated that the *Atbt3* mutant lines were more susceptible to salt stress than the WT. Moreover, over two dozen different protein domains were associated with the BTB, five of which (MATH, Kelch, NPH3, ion transport, and the zF domains) were much more frequent than the others [44]. Furthermore, the combination of the BTB domain with the TAZ domain is only observed in plants [45]. 

The other motif was NON-PHOTOTROPIC HYPOCOTYL 3 (NPH3), which was present in OsBTBN3 and OsBTBN7. The BTB–NPH3 proteins, also called NPH3/RPT2-like (NRLs) proteins, are plant-specific BTB/POZ proteins18. NRLs contain an N-terminal BTB domain and a C-terminal NON-PHOTOTROPHIC HYPOCOTYL 3 (NPH3) domain; some members contain an additional C-terminal coiled-coil domain [46]. NPH3, a BTB NPH3 family member in *Arabidopsis*, functions as a CRL3 substrate adaptor and regulates the ubiquitylation of phototropin1 (phot1) in response to different blue-light intensities. The dephosphorylation of NPH3 when stimulated by blue light may also be crucial for phot1-dependent phototropism [47]. Proteins containing NPH3 are abundantly localized in the plasma membrane and interact with phototropins and blue light receptor kinases [48]. 

Evidence from a putative cis-element analysis and PPI showed that *OsBTBZ1* and *OsBTBZ2* are related to the stress response. CARM1, GCN5, and OS07T0626600-01 are involved in post-translational modifications such as phosphorylation, glycosylation, acetylation, succinylation, carbonylation, S-nitrosylation, and Tyr-nitration, which can alter the epigenetic status of plants [49]. CARM1 histone-arginine methyltransferase, methylates (mono- and asymmetric dimethylation), and the guanidino nitrogens of arginyl residues in several proteins are involved in DNA packaging, transcription regulation, and mRNA stability. They can also be recruited to the promoters upon gene activation to methylate histone H3 and activate transcription via chromatin remodeling. Protein arginine methyltransferases (PRMTs), which are equivalent to CARM1 in mammals, were reported to be related to salt tolerance in *Arabidopsis*. Prmt5 and prmt4a;4b *Arabidopsis* mutants display an alteration in salt-stress tolerance [50]. Both OsBTBZ1 and OsBTBZ2 were predicted to directly interact with protein arginine methyltransferases, suggesting that both OsBTBZs are involved in salt-stress tolerance. The histone acetyltransferase GCN5 functions in acetylation of histone H3, which provides a specific tag for epigenetic transcriptional activation. GCN5 operates in concert with certain DNA-binding transcriptional activators that act via the formation of large multiprotein complexes to modify chromatin. Zheng et al. [51] reported that GCN5 plays an important role in cell wall integrity and salt tolerance in *Arabidopsis*. The wheat *TaGCN5* gene can complement the *Atgcn5* mutation, leading to the restoration of the salt-tolerant phenotype in the mutant line. This information suggested that OsBTBZ1 and OsBTBZ2 could be involved in chromatin remodeling and epigenetic regulation in the salt-tolerant phenotype in rice. It was also reported that the histone deacetylase, HDA710, regulated salt tolerance in rice via ABA signaling [52]. This is consistent with our results in this research that the complementation of *OsBTBZ1* in the *Atbt3* mutant background could restore the susceptible phenotypes of the *Atbt3* mutant under ABA treatment (Figure 7, Figure 8 and Figure 9). OS07T0626600-01 is a putative MYST-like histone acetyltransferase 1 (histone acetyltransferase), which may be involved in transcriptional activation. The involvement of regulation via acetylation and chromatin remodeling is consistent with an earlier report, which stated that the acetylation levels of histone H3 at K9 in maize increase during salt stress [53]. OS01T0884500-01 is another protein in the PPI of OsBTBZ1 that belongs to a group of protein-like zF (CCCH-type). In general, zinc fingers are of the C2H2-type or CCCC-type, being grouped by the configuration of cysteine and histidine. The less prevalent CCCH zF proteins are crucial for controlling plant stress responses [54].

OsBTBZ1 is also associated with OS03T0216600-01, a defense response protein elicited by PAMPs. This suggests an interaction between the responses to biotic and abiotic stresses. OsJ_06167, a putative WRKY transcription factor (TF), is one of the TFs that are involved in many biotic and abiotic stress regulations. Epigenetic, retrograde, and proteasome-mediated regulations enable WRKYs to attain dynamic cellular homeostatic reprogramming [55]. Li et al. [56] reported that the overexpression of *SpWRKY1* in tobacco resulted in enhanced salt and drought stress tolerance by reducing lipid peroxidation, enhancing antioxidant enzyme activity, and maintaining photosynthesis. The promoter of *OsBTBZ1* also had a motif that is associated with the WRKY protein. WRKY71OS and WBOXATNPR1 are involved in various stress responses and are mediated by gibberellic acid, ABA, and salicylic acid [57]. In rice, WRKY13 binds to multiple cis-elements to regulate abiotic and biotic stress [58]. OsJ_25984 belongs to the protein kinase superfamily. Kinases are necessary for signal transduction in many aspects of cellular regulation and metabolism. The regulation of plant growth and development and the plant’s responses to stress conditions involve protein kinases such as mitogen-activated protein kinase cascades, receptor-like kinases, sucrose nonfermenting1-related protein kinases, and calcium-dependent protein kinases [59]. 

The putative cis-element analysis also showed interesting results, summarizing the function of *OsBTBZ1*. The ABA-responsive element (ABRE) is the most conserved cis-element in plants [60,61]. The ABRE cis-elements control the transcriptional regulation of several genes in response to cytosolic Ca^2+^ [62]. CAMTA12 enhances drought tolerance in soybean and *Arabidopsis* by binding to the ABRE cis-elements [63]. Many TFs regulate both abiotic and biotic stress, such as cold, drought, heat, and salinity, by binding to different cis-elements [64]. The OsBTBZ1 promoter contains putative MYB-binding sites that are attached to the MYB transcription factor. MYB cis-elements, including MYBCORE, and MYB1AT, were detected in the promoter regions of *OsBTBZ1*, *OsBTBZ2*, *OsBTBN3*, and *OsBTBN7*; however, the number of elements varied. They play essential roles in the regulation of many genes related to biotic and abiotic stresses [65]. ABRE and MYC play important roles in the ABA-induced activation of biotic and abiotic genes [66]. MYC also responds to drought stress [28]. For example, AtMYC2 and AtMYB2 specifically interact with MYB recognition sites to regulate the ABA genes related to biosynthesis and signaling [67]. In another study, a chromatin immunoprecipitation assay and effector-reporter coexpression assays of *Nicotiana tabacum* confirmed the relationship of *MYB* and *WRKY* cis-elements with the promoters of peroxidase, superoxide dismutase, and phenylalanine ammonia-lyase to regulate abiotic stress [68]. Additionally, MYB- and WRKY-related cis-elements were found to regulate the transcription of auxin-regulated genes [69]. Similarly, the dehydration-responsive element (*DRE*) is also an essential cis-element that regulates the drought response and other abiotic stresses in plants. DREB-responsive genes are regulated by the *DREB* cis-elements to mediate stress responses in plants [70,71]. In banana plants, DREBs mitigate heat and drought stress [71], whereas in soybean plants, the cis-elements of *DREBs* are important for proline accumulation to mediate the plant’s response to salt stress [72].

*OsBTBZ1* was selected to confirm its role in salt tolerance by generating the revertant lines REV1 and REV2 in the *Atbt3 Arabidopsis* mutant. Upon adding 150 mM of mannitol, the expression of the *OsBTBZ1* gene could increase the tolerance of the transgenic lines to salt stress more than the mutant line, but not to drought stress (Figure 7, Figure 8 and Figure 9). This suggested that the function of *OsBTBZ1* was more specific to the salt stress response than to osmotic stress. Phenotypic complementation of the mutant line by the expression of *OsBTBZ1* was also observed under the ABA treatment, suggesting that the mechanism via the ABA-dependent pathway is involved in *OsBTBZ1* functioning, which is consistent with the model proposed by Chutimanukul et al. [31].

## 4. Materials and Methods

### 4.1. Transcriptome Analysis

A transcriptome study was conducted during the seedling and booting stages of the CSSL16 line. It is a chromosome substitution salt-tolerant line with a ‘KDML105’ genetic background from BC_5_F_4_, originally taken from a cross between ‘KDML105’ and DH105, which was selected as an abiotic stress-tolerant double haploid line. After the cross, F_1_ progeny was backcrossed to ‘KDML105’ for 5 generations and then, self-fertilized to create BC_5_F_4_. Marker-assisted selection was used to select those CSSLs with the abiotic stress-tolerant regions from DH105 [31,32]. Transcriptomic data were retrieved from the database of the National Center for Biotechnology Information under the BioProject IDs PRJNA507040 and PRJNA659381 [31]. Briefly, the CSSL16 line plants were grown under normal and salt stress conditions (75 mM of NaCl treatment). The total RNA from 21-day-old seedling leaves was extracted after 0 and 48 h of salt stress treatment, whereas the total RNA from the flag and second leaves at the booting stage was extracted at 0 h and 72 h after salt stress, respectively, using a plant RNA purification reagent (Invitrogen, USA). Genomic DNA was extracted using DNase I (Invitrogen, Waltham, MA, USA). A KAPA stranded RNA-Seq library preparation kit (Illumina, San Diego, CA, USA) was used to synthesize the cDNA libraries, which were sequenced using Illumina next-generation sequencing (Illumina, USA). The differentially expressed genes were identified using the DESeq tool, version 1.24.0 [73]. Genes with significantly different expressions were considered those with a *p*-value of < 0.01. The PC, sequencing data matrix, and box plot showing the quality of the transcriptome data are shown in Appendix A.

### 4.2. Phylogenetic Analysis and an In Silico Analysis of BTB Proteins in Oryza sativa

The amino acid sequences from the Rice Genome Annotation Project database were subjected to a motif search (https://www.genome.jp/tools/motif/ accessed on 10 October 2022) to predict the protein motifs [74]. All BTB proteins were retrieved from the Phytozome database, available at https://phytozome.jgi.doe.gov/pz/portal.html [75], accessed on 25 October 2022. Orthologous BTB proteins from *A. thaliana* were retrieved from the TAIR database (https://www.arabidopsis.org/ accessed on 25 October 2022). In total, 209 proteins (182 from rice and 27 from *A. thaliana*) containing the BTB domain were used to construct a phylogenetic tree. All BTB protein sequences were aligned in MEGAX, while for the maximum likelihood, the Jones–Taylor–Thornton (JTT) method was employed for phylogenetic tree construction. The phylogenetic tree thus obtained from MEGAX was visualized using iTOL, which is available at https://itol.embl.de/ [76] accessed on 3 November 2022. All the *BTB* genes obtained from the Phytozome database were subsequently subjected to the Oryzabase-integrated rice science database http://viewer.shigen.info/oryzavw/maptool/MapTool.do [77] (accessed on 28 October 2022).

### 4.3. Putative Promoter Analysis

A promoter analysis of *BTBZ1* was carried out on a sequence retrieved from the Phytozome database (https://phytozome-next.jgi.doe.gov/, accessed on 1 November 2022) [37], from 0 to −2000 bps, and entered into the New Plant cis-acting regulatory DNA elements (New PLACE) website (https://www.dna.affrc.go.jp/PLACE/?action=newplace, accessed on 7 November 2023). The positions of stress-related cis-regulatory elements were visualized using the TBtools software, and the functions of these elements were mainly obtained from the New PLACE database and the published literature [78].

### 4.4. Protein-Protein Interaction (PPI) Based on the STRING Database

The PPI was predicted via STRING [39]. The LOC_Os01g66890 (OsBTBZ1), LOC_Os01g68020 (OsBTBZ2), LOC_Os02g38120 (OsBTBN3), and LOC_Os03g41350 (OsBTBN7) proteins have been named according to the Rice Genome Annotation project (http://rice.uga.edu, accessed on 20 November 2022). However, the STRING website recognizes the Rice Annotation Project Database (RAP-DB) ID. To facilitate this analysis, the locus numbers based on the Rice Genome Annotation Project were converted into RAP-DB IDs using the following tool (https://rapdb.dna.affrc.go.jp/tools/converter/run, accessed on 20 November 2022). The RAP-DB id of four *BTB* genes are as follows: *Os01g0893400* (*OsBTBZ1*), *Os01g0908200* (*OsBTBZ2*), *Os02g0594700* (*OsBTBN3*), and *Os03g0609800* (*OsBTBN7*). The interactome was produced using a full-string network based on automated text mining, high-throughput experiments, prior stored databases, and co-expression sources. False-positive and false-negative results were reduced using a high confidence score (0.700). The other parameters were set to the default values. Enrichment detection was used to predict a network that covered all the mapped proteins and their interconnections.

### 4.5. Detection of OsBTBZ1 Gene Expression

A quantitative real-time polymerase chain reaction (qRT-PCR) was used to investigate the expression profile of *OsBTBZ1* in various tissues of Nipponbare rice varieties under normal conditions. Various tissues were selected at specific time points, as follows. In 15-day-old seedlings, the leaf blade and leaf sheath of the first fully expanded leaf, the second-youngest leaf, and the oldest leaf, including the root tissues, were collected for the gene expression study. Then, in 30-day-old plants, only the leaf blade of a fully expanded leaf was collected. At the reproductive stage, the leaf blade and leaf sheath of the flag leaf, the panicle, and the spikelets were collected to investigate gene expression. Three biological replicates were used for each analysis. Total RNA from all tissues was extracted using the GENEzolTM reagent GZR100, following the manufacturer’s protocol, and then treated with DNase I. cDNA synthesis was performed using the Accupower RT premix (Bioneer Inc., Alameda, USA) and using oligoDT(T)18 as a primer to produce 40 ng/μL cDNA. The synthesized cDNA was used as a template and qRT-PCR was conducted on a Luna Universal qPCR master mix M3003L (New England Biolabs Inc., Ipswich, MA, USA). The qRT-PCR conditions were: 95 °C for 60 s, followed by 39 cycles at 95 °C for 15 s, 61.5 °C for 30 s, and 95 °C for 5 s; furthermore, the melt curve and plate read were at 60–94 °C, along with an increase in temperature of 5 °C per 5 s. A qRT-PCR was conducted in triplicate for each sample. The negative control was performed without a template and those reactions containing OsEF-1α primers (Table 2) were used as the internal reference genes [79]. The coding sequence obtained from the rice genome annotation database [37] was used to design primers that were specific to *OsBTBZ1* (Table 2). The average cycle threshold (Cq) values of the gene were normalized with the level of the *OsEF-1α* reference gene in the same sample and were then used to measure the relative gene expression using the method described by Pfaffl [80]. Gene expression was analyzed using the relative expression levels. The statistical program SPSS was used to conduct an analysis of variance (ANOVA, *p* < 0.05) and the means were compared using Duncan’s multiple range test.

### 4.6. Generation of Complementation and Over-Expression of Arabidopsis Lines with the OsBTBZ1 Gene 

A full-length *OsBTBZ1* cDNA clone, J023077N08, obtained from the NARO DNA Bank was cloned into the *Escherichia coli* DH5α strain and cultured onto LB semisolid medium (10 g tryptone, 5 g yeast extract, 10 g NaCl, and 7.5 g agar per liter of water) along with 100 µg/mL of ampicillin. Single colonies were selected and then cultured into the LB broth with ampicillin for 16 h at 37 °C and 200 rpm. The plasmid was extracted using the Presto mini plasmid kit according to the manufacturer’s protocol (Geneaid, Taiwan), and the sequence of the inserted fragment was determined using the M13 (-20) forward primer to validate the correct sequence of the *OsBTBZ1* gene.

The expression vector was constructed using the Gateway system. The *OsBTBZ1* gene was added with the CACC adaptor in 5′ ends, inserted into the pENTR D-TOPO plasmid (Thermo Fisher Scientific, Waltham, MA, USA) as a donor vector, and then cloned to the TOP10 *E. coli* strain using the heat shock method. The cells were then plated on an LB medium, supplemented with kanamycin (50 µg/mL), and were subsequently incubated at 37 °C overnight. The correct sequence of the plasmid host was utilized in the LR clonase reaction to switch the *OsBTBZ1* gene from the donor vector pENTR to the pGWB512 and pGWB505 plasmids as the destination vectors. The destination vector was transferred to the *E. coli* strain DH5α and cultured on LB medium, supplemented with 50 µg/mL spectinomycin. Plasmids with the correct sequence were transferred to *Agrobacterium tumefaciens* GV3101-competent cells using the cold-shock method. To identify colonies with the inserted fragment, a polymerase chain reaction (PCR) was performed using the *CaMV35S* forward primer and the reverse primer in the *OsBTBZ1* gene to identify the correct clone for plant transformation.

### 4.7. Transformation of A. thaliana

An *OsBTBZ1* cDNA in the pGWB512 construct was inserted into *Arabidopsis* plants using the floral dip transformation [81]. The resulting T_1_ plants were positive for the *OsBTBZ1* gene and were grown in soil to obtain T_2_ seeds, which were acquired from each transgenic line and then germinated on Murashige and Skoog (MS) medium, supplemented with 25 mg/L hygromycin. The 3:1 segregation ratio of resistance: sensitivity to hygromycin was determined to identify transgenic lines with a single insertion. The selected lines were then grown in soil to obtain homozygous T_3_ seeds, which were used for further characterization [82].

### 4.8. Subcellular Localization in Onion Inner Epidermal Cells

The agroinfiltration of onion (*Allium cepa*) inner epidermal cells was performed to observe the subcellular localization of OsBTBZ1. The *Agrobacterium* GV3101 harboring the *CaMV35S::OsBTBZ1-GFP* construct in pGWB505 was cultured into 5 mL LB, supplemented with the appropriate antibiotics (50 µg/mL spectinomycin, 25 µg/mL rifampicin, and 50 µg/mL gentamicin) at 28 °C for 1 d. Later, 250 µL of the culture was inoculated in 25 mL LB, supplemented with 10 mM MES (pH 5.6, 100 µM acetosyringone, and antibiotics), and grown at 28 °C to obtain an optical density (OD) at 600 of 0.8. Subsequently, the culture was centrifuged at 5000 rpm for 5 min. The cell pellet was resuspended in MMA liquid medium (10 mM MgCl_2_, 10 mM MES (pH 5.6), and 100 µM acetosyringone) to a final OD600 of 0.8–1.0 [83]. The mixture was incubated at room temperature (25–26 °C) for 3 h and then infiltrated using a 5 mL needleless syringe, with only 500 µL injected per spot in the onion epidermal cells [84]. The onions were incubated in dim light or in dark conditions at 22 °C under high humidity for 48 h. The *Agrobacterium* GV3101, containing *CaMV35S::GFP*, was used as a control and was treated in the same manner. The onion epidermal cell layers were cut into 1 × 1 cm^2^ squares, peeled, and transferred directly to glass slides. Subsequently, 40 µL of 1 µg/mL 4′,6-diamidino-2-phenylindole, or DAPI dye was added to the epidermal cell sections. The green fluorescence protein (GFP) signal was observed under a Zeiss microscope (ZEISS Axio 10, Göttingen, Germany) at an excitation wavelength of 488 nm.

### 4.9. Evaluation of the Effect of OsBTBZ1 Gene Expression in Transgenic Arabidopsis Lines

The WT *Arabidopsis*, *Atbt3* mutants, two homozygous complemented lines, REV1 and REV2, and two ectopic expression lines in the WT background, OE1 and OE2, were used for phenotyping. The experiment was performed using a completely randomized design with three replicates. The seeds were germinated by subjecting the seeds to the 0.5× MS medium as a control or 0.5× MS medium supplemented with 150 mM NaCl, 1 µM ABA, or 150 mM mannitol. Germinating seeds were recorded from 0–11 d to calculate the germination rate. 

To elucidate the growth responses, 7-day-old *Arabidopsis* seedlings of each line were grown in 0.5× MS medium and were then subjected to four treatments: 0.5× MS medium as a control, 0.5× MS medium, supplemented with 150 mM NaCl, 0.5× MS medium supplemented with 1 µM ABA, and 0.5× MS medium supplemented with 150 mM mannitol, to investigate the impact on fresh weight, root length, and pigment contents. All parameters were measured 0–6 d after initiating the treatments. Fresh weight was calculated as the weight per plant, while the root length was calculated as the change in root length during the experimental period. The pigments, chlorophyll *a*, chlorophyll *b*, and total carotenoid contents, were determined according to the procedures described by Wellburn [85]. All experiments were performed in at least three replicates. An ANOVA was performed using SPSS Statistical Software version 23 (IBM Corp., Armonk, USA), followed by Duncan’s multiple-range test, to compare the means of each parameter.

## 5. Conclusions

Our study indicates a clear role for *OsBTBZ1* in salt tolerance in *Arabidopsis*. A role in salt tolerance in rice is consistent with the higher expression of this gene in the salt-tolerant line, CSSL16, compared to the original genetic background, ‘KDML105’ rice [31]. It is also consistent with the location of this gene on the salt tolerance QTL on chromosome 1 [31,86]. Therefore, this study can support the use of this gene and QTL for the improvement of salt tolerance in rice. 

## Figures and Tables

**Figure 1 ijms-24-14483-f001:**
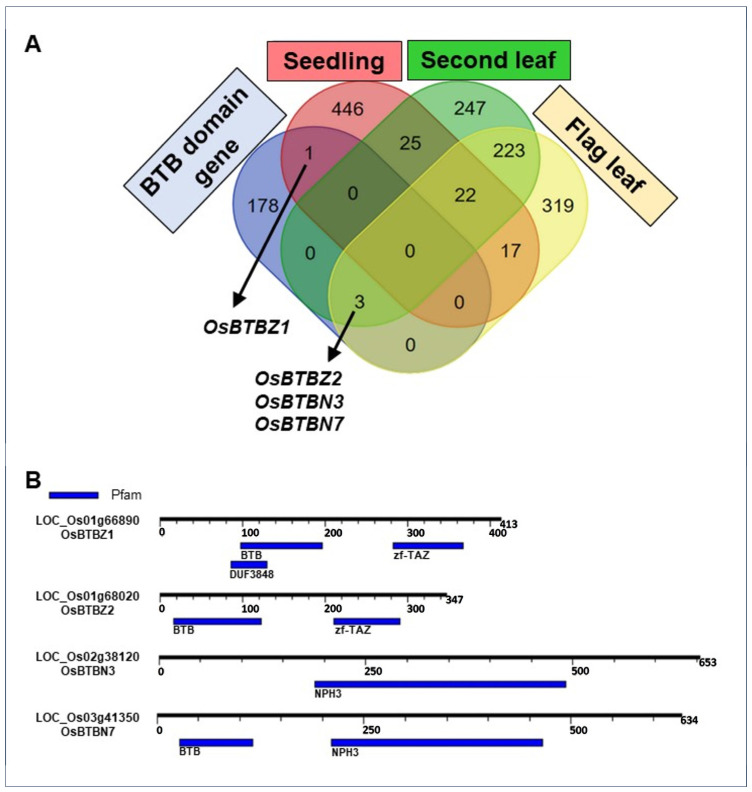
Venn diagram showing the intersection of DEseq data with the BTB protein in *O. sativa* (**A**). BTB, zf-TAZ, and NPH3 domains are present in the differentially expressed BTB genes under salt stress conditions, namely, OsBTBZ1, OsBTBZ2, OsBTBN3, and OsBTBN7 (**B**).

**Figure 2 ijms-24-14483-f002:**
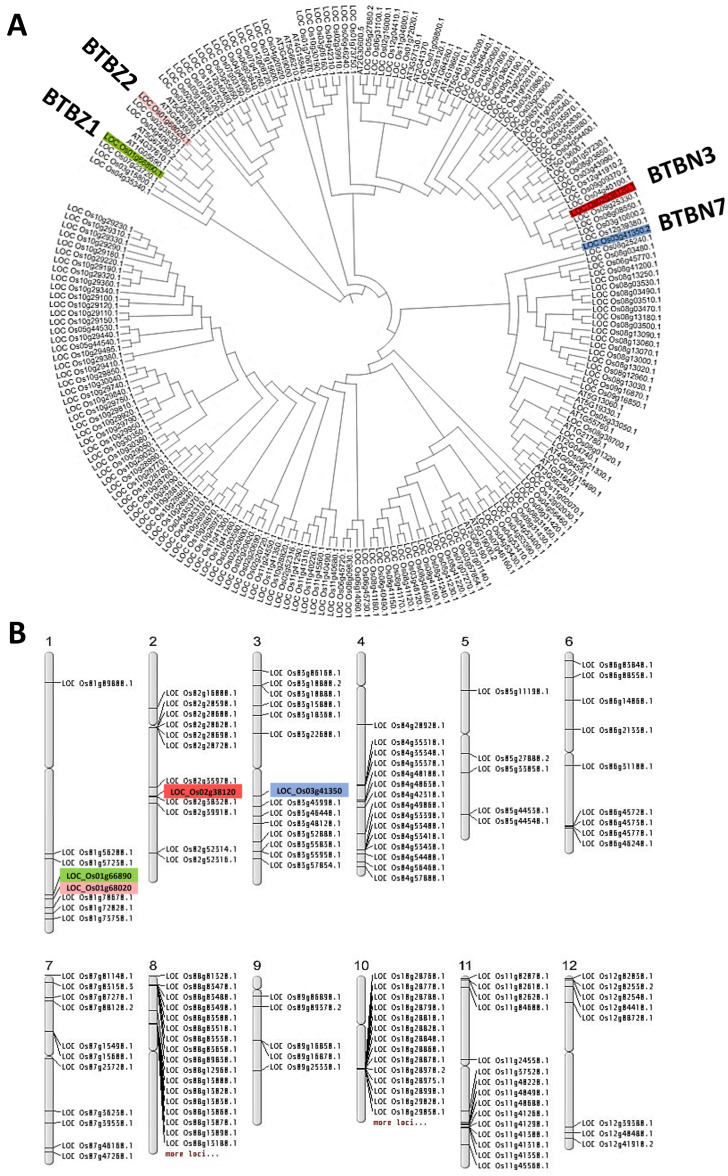
Phylogenetic tree and the position of *BTB genes* in the rice chromosome. (**A**) Maximum likelihood phylogenetic tree. The tree was constructed using the amino acid sequences of 27 *BTB* genes in *Arabidopsis* and 182 *BTB* genes in rice. *LOC_Os01g66890* (*BTBZ1*) was marked, along with *LOC_Os01g68020*, *LOC_Os02g38120*, and *LOC_Os03g41350*. (**B**) The chromosomal location shows the distribution of the *BTB* gene family in the 12 rice chromosomes.

**Figure 3 ijms-24-14483-f003:**
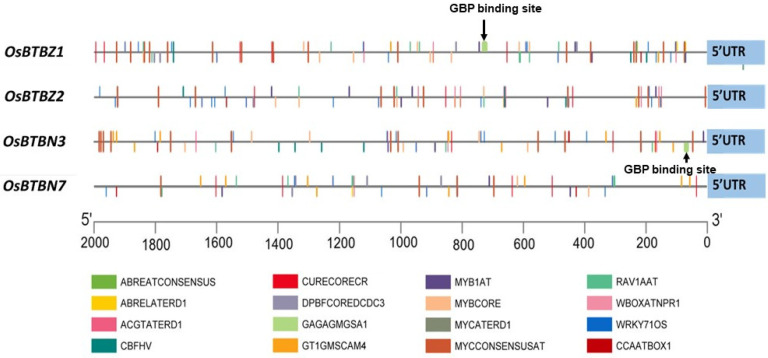
Prediction of the cis-elements related to stress in OsBTBZ1, *OsBTBZ2*, *OsBTBN3*, and the *OsBTBN7* promoter.

**Figure 4 ijms-24-14483-f004:**
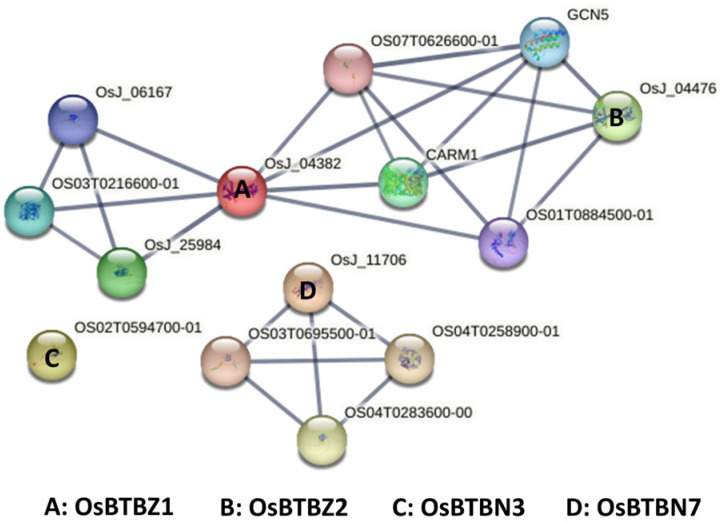
The protein-protein interaction (PPI) network assembly of LOC_Os01g66890 (A: OsBTBZ1), LOC_Os01g68020 (B: OsBTBZ2), LOC_Os02g38120 (C: OsBTBN3), and LOC_Os03g41350 (D: OsBTBN7) proteins, identified using STRING. The lines represent a high confidence PPI score of 0.7.

**Figure 5 ijms-24-14483-f005:**
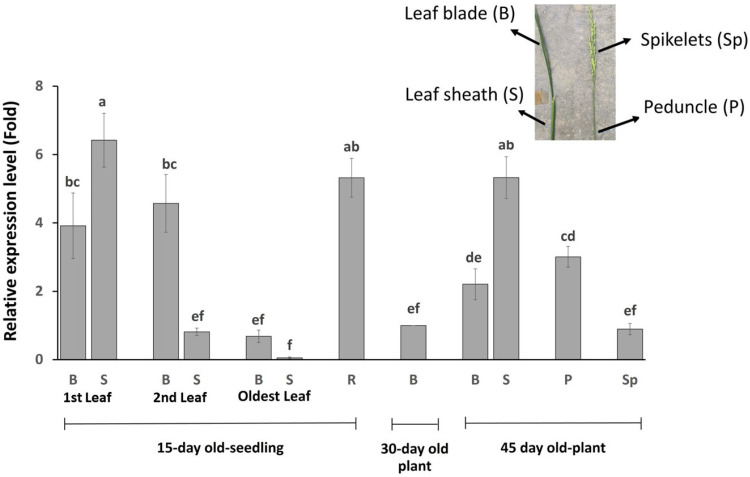
*OsBTBZ1* gene expression in different tissues of “Nipponbare” rice. B = leaf blade; S = leaf sheath; P = peduncle; R = root; Sp = spikelet. The different lowercase letters above the bars mean the significant difference between means analyzed by Duncan’s multiple range test at *p* < 0.05.

**Figure 6 ijms-24-14483-f006:**
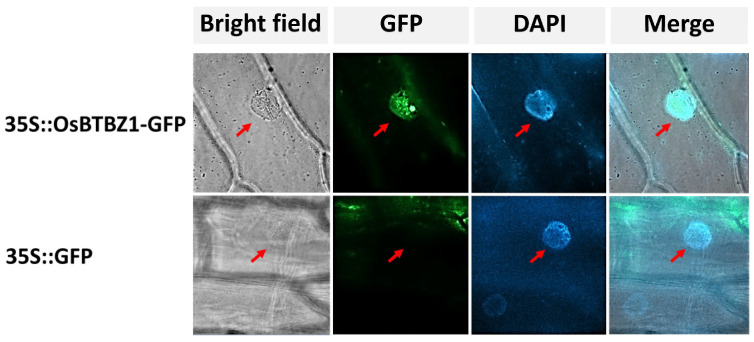
The subcellular localization of OsBTBZ1-GFP in onion (*Allium cepa*) epidermal cells. Red arrows point to the location of the nucleus.

**Figure 7 ijms-24-14483-f007:**
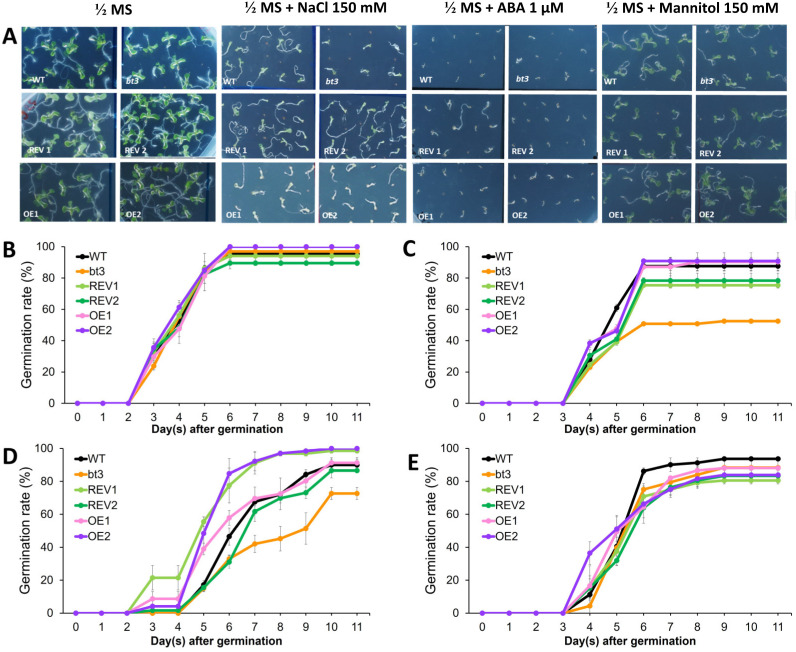
Germination tests of the wild-type (WT), mutant, and transgenic lines in several different media. *(***A**) The lines of WT, bt3 mutant, REV1, REV2, OE1, and OE2 in ½ MS basal medium as a control, medium with NaCl 150 mM, ABA 1µM, and mannitol 150 mM (line mark = 1 cm). *(***B***)* Germination curve in the control medium (½ MS medium). *(***C**) Germination curve in the ½ MS medium containing 150 mM NaCl. *(***D**) Germination curve in the ½ MS medium containing 1 µM ABA. *(***E**) Germination curve in ½ MS medium containing 150 mM mannitol.

**Figure 8 ijms-24-14483-f008:**
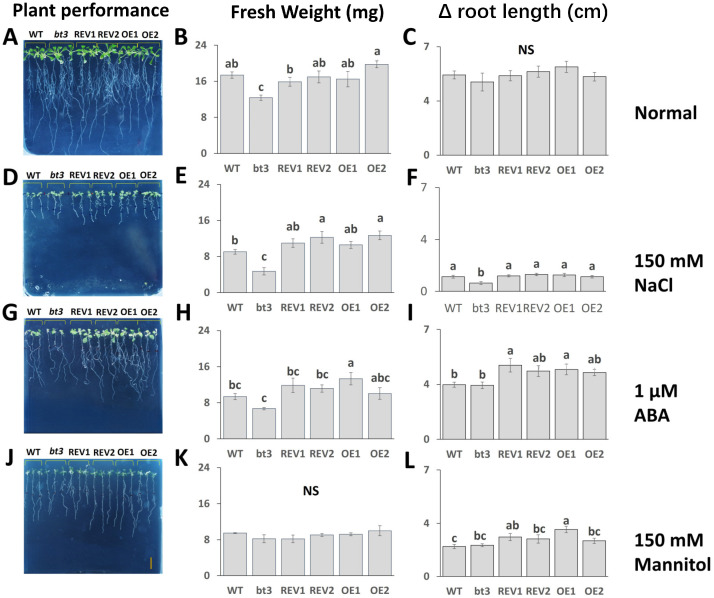
Growth (**A**,**D**,**G**,**J**) WT, bt3 mutant, and transgenic lines of 7-day-old *Arabidopsis* seedlings in various conditions: normal (control), supplemented with 150 mM NaCl, 1 mM ABA, or 150 mM mannitol on day 6 of stress exposure. (**B**,**E**,**H**,**K**): fresh weight of plants; (**C**,**F**,**I**,**L**): Δ root length of plants. The different letters above the bars represent the significant difference in means at *p* < 0.05 and NS represents no significant difference.

**Figure 9 ijms-24-14483-f009:**
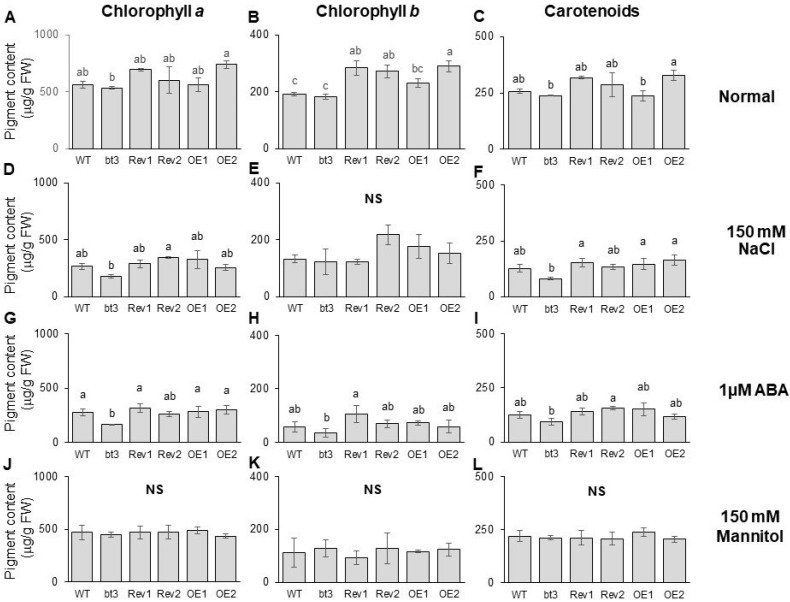
Contents of pigments, namely, Chl a, (**A**,**D**,**G**,**J**), Chl b, (**B**,**E**,**H**,**K**), and carotenoids (**C**,**F**,**I**,**L**) of 13-day-old Arabidopsis seedlings treated with 150 mM NaCl, 1 mM ABA, or 150 mM mannitol for 6 d. The data were collected from the WT and bt3 complemented lines with the OsBTBZ1 gene in the bt3 mutant background (Rev1 and Rev2) and the OsBTBZ1 ectopic expression line with a WT background (OE1 and OE2). The different letters above the bars represent the significant difference in means at *p* < 0.05 and NS represents no significant difference.

**Table 1 ijms-24-14483-t001:** The number of Stress-related cis-elements detected in *OsBTBZ1*, *OsBTBZ2*, *OsBTBN3*, and *OsBTBN7*.

Factor or Site Name	Signal Sequence	*OsBTBZ1*		*OsBTBZ2*		*OsBTBN3*		*OsBTBN7*		Functions
		Strand (+)	Strand (−)	Strand (+)	Strand (−)	Strand (+)	Strand (−)	Strand (+)	Strand (−)	
ABREATCONSENSUS	YACGTGGC	1	-	-	-	-	-	-	-	ABA-responsive element
ABRELATERD1	ACGTG	3	3	-	6	1	1	-	1	*ERD*-related gene (early responsive to dehydration)
ACGTATERD1	ACGT	5	5	8	8	4	4	1	1	Abiotic stresses (drought, salt); response to light
CBFHV	RYCGAC	1	3	2	2	-	4	-	-	Dehydration-responsive element (DRE) binding proteins (DREBs)
CCAATBOX1	CCAAT	-	2	-	1	1	1	-	2	Heat-shock element
CURECORECR	GTAC	5	5	3	3	3	3	5	5	Oxygen-responsive element
DPBFCOREDCDC3	ACACNNG	4	3	1	3	1	1	3	2	Dehydration and ABA response
GAGAGMGSA1	GAGAGAGAGAGAGAGAGA	2	-	-	-	-	1	-	-	GAGA binding protein (GBP) binding site
GT1GMSCAM4	GAAAAA	1	1	-	1	5	3	6	2	Plays a role in pathogen- and salt-induced SCaM-4 gene expression
MYB1AT	WAACCA	2	1	4	2	1	1	1	2	Element for the dehydration-responsive gene in *Arabidopsis*
MYBCORE	CNGTTR	5	6	2	4	3	3	1	1	Responsive to water stress, induced by dehydration stress
MYCATERD1	CATGTG	1	-	2	-	1	2	2	-	Necessary for *ERD1* expression, the binding site of the NAC protein
MYCCONSENSUSAT	CANNTG	16	16	10	10	12	12	4	4	Element for the dehydration-responsive gene in *Arabidopsis*
RAV1AAT	CAACA	5	4	2	1	-	3	6	1	Rosette leaves and the root-specific element, growth, and development, hormonal regulation brassinosteroid (BR); metabolism
WBOXATNPR1	TTGAC	2	2	1	3	1	1	2	1	Salicylic acid-induced WRKY DNA binding proteins
WRKY71OS	TGAC	8	6	3	11	8	2	3	4	Involved in gibberellic acid (GA), ABA-mediated pathways, and pathogen-related protein (PR)

**Table 2 ijms-24-14483-t002:** Primer sequence for qRT-PCR to detect gene expression.

Name	Sequence 5′→3′	T_m_ (°C)
qPCR_OsBTBZ1_FW	TTCCTGCCTGCAAGGGCATC	63
qPCR_OsBTBZ1_REV	TCCTTGAAATGCCTACAGAGGGG	60
qPCR_OsEF1α_FW	ATGGTTGTGGAGACCTTC	53
qPCR_OsEF1α_REV	TCACCTTGGCACCGGTTG	60

## Data Availability

The transcriptome sequences are available in the NCBI biotechnology information database under Bioproject IDs: prjna507040 and prjna659381.

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
