# Peer review of "OsBTBZ1 Confers Salt Stress Tolerance in Arabidopsis thaliana"

_ijms, 2023, doi:10.3390/ijms241914483_

Round 1

Reviewer 1 Report

Saputro et al. demonstrated the rescuing capability of the OsBTBZ1 gene from rice in the Arabidopsis bt3 mutant, which exhibited heightened susceptibility to NaCl and ABA. The expression of OsBTBZ1 was found across all plant tissues, with a particularly strong presence in younger leaf sheaths. The study provided preliminary insights into the functions of OsBTBZ1. However, in order to enhance the clarity and logical progression of the manuscript, the authors are advised to offer a more comprehensive rationale for conducting their research. For instance, clarification regarding the choice of presenting cis-elements and the Protein-Protein Interaction (PPI) network in sections 2.1 and 2.3 is necessary, as the current presentation appears perplexing in this version. Additionally, concluding statements are absent in several sections, including 2.6 and 2.7, warranting attention. It is also essential to ensure the inclusion of pertinent references that appear to be missing in various parts of the manuscript, such as lines 54-55 and 61-63. The manuscript holds promise but can be further refined for clarity, coherence, and comprehensive referencing. Addressing the aforementioned points will contribute to a more robust and comprehensible publication.

Several comments have been noted:

The term "Arabidopsis" should be italicized throughout the manuscript. Please review and make the necessary corrections.

Line 19: Provide the full name of DEseq for clarity.

Lines 51-54: Condense these sentences to eliminate redundancy and enhance conciseness.

In the introduction section, it is recommended that the authors provide an overview of the advancements in salt stress tolerance-related studies in rice to establish a clear research background.

Line 82: Briefly describe the functions of the JTT model that is used here.

The resolution of Figure 2A and 2B should be improved to ensure the clarity of details.

Line 115: Define the full name of PPI.

Line 103: Should cite Table 1 here.

In Figure 5, clarify the meaning of the "R" displayed.

The fluorescence signal of 35S::GFP in Figure 6 requires enhancement for improved visibility.

Line 168: It would greatly enhance the manuscript if the authors could experimentally verify the transcriptional factor activity of OsBTBZ1 through techniques such as Electrophoretic Mobility Shift Assay (EMSA).

Line 222: Provide an explanation for conducting seeds germination in the presence of mannitol in this context.

Line 227: The subtitle should be section 2.6.

Figure 9: panels E, J, K, and L, should include statistical results for completeness.

Reviewer 2 Report

Congratulations to the authors for their work on the article "OsBTBZ1 Confers Salt Stress Tolerance in Arabidopsis thaliana," submitted to the International Journal of Molecular Sciences. The manuscript addresses an important topic related to salt stress tolerance mechanisms in plants. While the manuscript is well-structured and provides valuable insights, I have a few minor comments and suggestions to improve the clarity and completeness of the content.

Minor Comments:

Introduction:

The introduction is short it should provide a brief overview of the study's context of salt tolerance genes and their function similar to OsBTBZ1 with citing previous studies.

Results:

In the beginning the section regarding overall results of transcriptome analysis, quality of library preparation, PCA plot etc.. and detailed explanation of the rationale behind choosing the OsBTBZ1 gene domain for venn diagram preparation and the criteria for selecting specific genes and pathways for analysis.

Also there is no experiment set up explanation about why you choose seedling, second leaf and falg leaf  and transcriptome data comparison against BTB domain.

Regarding Figure 5, the significance calculation for relative expression levels among samples should be calculated and added.

The overall figure quality should be clearer with large font size, also in tables.

Discussion:

I recommend more extensive discussion on the interaction between OsBTBZ1 and salt stress-related genes, as well as references to similar studies that elucidate related mechanisms or methodologies. Providing a more comprehensive context will strengthen the interpretation of your findings.

Supplementary Materials:

it is suggested to include qualitative information about the transcriptome analysis that include PC and coexpression analysis, a sequencing data matrix, and box plots to showcase the quality of the sequencing analysis.
